# Temporal alignment and latent Gaussian process factor inference in population spike trains

**Lea Duncker & Maneesh Sahani**
Gatsby Computational Neuroscience Unit
University College London
London, W1T 4JG
{duncker,maneesh}@gatsby.ucl.ac.uk

## Abstract

We introduce a novel scalable approach to identifying common latent structure in neural population spike-trains, which allows for variability both in the trajectory and in the rate of progression of the underlying computation. Our approach is based on shared latent Gaussian processes (GPs) which are combined linearly, as in the Gaussian Process Factor Analysis (GPFA) algorithm. We extend GPFA to handle unbinned spike-train data by incorporating a continuous time point-process likelihood model, achieving scalability with a sparse variational approximation. Shared variability is separated into terms that express condition dependence, as well as trial-to-trial variation in trajectories. Finally, we introduce a nested GP formulation to capture variability in the rate of evolution along the trajectory. We show that the new method learns to recover latent trajectories in synthetic data, and can accurately identify the trial-to-trial timing of movement-related parameters from motor cortical data without any supervision.

## 1 Introduction

Many computations in the brain are thought to be implemented by the dynamical evolution of activity distributed across large neural populations. As simultaneous recordings of population activity have become more common, the need has grown for analytic methods that can identify these dynamical computational variables from the data. Where the computation is tightly coupled to an externally measurable covariate — a stimulus or a movement, perhaps — such identification is a simple matter of exploiting linear or non-linear regression to describe a population encoding or decoding model. However, when aspects of the computation are likely to reflect internal mental states, which may vary substantially even when external covariates remain constant, the relevant variables must be uncovered from neural data alone, most typically using a latent variable model [1–6].

However, most such methods fail to account properly for at least one key form of dynamical variability — trial-to-trial differences in the timing of the computation. Such differences may be reflected in behavioural variability, for instance in reaction times or movement durations [7], or in varying relationships between external events and neural firing, for instance in sensory onset latencies [8]. In some cases manual alignment to salient external events or behavioural time-course may be used to reduce temporal misalignment [8, 9]. However, just as with variability in the trajectories themselves, temporal variations in purely internal states must ultimately be identified from neural data alone [10].

A particularly challenging problem is to build models that capture the variability *both* in the latent trajectories underlying spiking population activity, and in the time-course with which such trajectories are followed. Temporal misalignment in trajectories might be confused for variability in the trajectory itself; while genuine variability in that trajectory makes alignment more difficult. Indeed, previous work has considered these problems separately. Algorithms like dynamic time warping (DTW) [11]

treat the time-series as observed and try to estimate an optimal alignment of each series. DTW and related approaches may suffer in settings where observed data are noisy, though recent work has begun to explore more robust [12–15], or probabilistic alternatives for time-series alignment [16–18]. Furthermore, these approaches generally assume a Gaussian noise model and often make assumptions that restrict applications to non-conjugate observation models. Only recently have studies considered pairwise alignment of univariate [19–21], or multivariate point-processes [22]. Overall, simultaneous inference and temporal alignment of latent trajectories from multivariate point process observations – such as a set of spike-times of simultaneously recorded neurons – is a relatively unexplored area of research.

In this work, we develop a novel method to jointly infer shared latent structure directly from the spiking activity of neural populations, allowing for (and identifying) trial-to-trial variations in the timing of the latent processes. We make two main contributions.

First, we extend Gaussian Process Factor Analysis (GPFA) [2], an algorithm that has been successfully applied in the context of extracting time-varying low-dimensional latent structure from binned neural population activity on single trials, to directly model the point-process structure of spiking activity. We do so using a sparse variational approach, which both simplifies the adoption of a non-conjugate point-process likelihood, and significantly improves on the scalability of the GPFA algorithm.

Second, we combine the sparse variational GPFA model with a second Gaussian process in a nested model architecture. We exploit shared latent structure across trials, or groups of trials, in order to disentangle variation in timing from variation in trajectories. This extension allows us to infer latent time-warping functions that align trials in a purely unsupervised manner – that is, using population spiking activity alone with no behavioural covariates. We apply our method to simulated data and show that we can more accurately recover firing rate estimates than related methods. Using neural data from macaque monkeys performing a variable-delay centre-out reaching task, we demonstrate that the inferred alignment is behaviourally meaningful and predicts reaction times with high accuracy.

## 2 Gaussian Process Factor Analysis

Gaussian Process Factor Analysis (GPFA) is a method for inferring latent structure and single-trial trajectories in latent space that influence the firing of a population of neurons [2]. Temporal correlations in the high-dimensional neural population are modelled via a lower number of shared latent processes $x_k(\cdot)$, $k = 1, \ldots, K$, which linearly relate to a high-dimensional signal $\boldsymbol{h}(\cdot) \in \mathbb{R}^N$ in neural space. Thus, inter-trial variability in neural firing that is shared across the population is modelled via the evolution of the latent processes on a given trial.

In the GPFA model, a Gaussian process (GP) prior is placed over each latent process $x_k(\cdot)$, specified via a mean function $\mu_k(\cdot)$ and a covariance function, or kernel, $\kappa_k(\cdot, \cdot)$. The extent and nature of the temporal correlations is specified by $\kappa_k(\cdot, \cdot)$ and governed via hyperparameters.

The classic GPFA model in [2] considers regularly sampled observations of $\boldsymbol{h}(t)$ that are corrupted by axis-aligned Gaussian noise. Recent work has aimed to extend GPFA to a Poisson observation model [23]. Here, $\boldsymbol{h}(t)$ is related to a piecewise-constant firing rate of a Poisson process whose counting process is observed as spike-counts falling into time bins of a fixed width.

We can summarise the GPFA generative model for observations $y_n^{(r)}(t_i)$ of neuron $n$ on trial $r$ in a general way by writing

$$x_k^{(r)}(\cdot) \sim \mathcal{GP}(\mu_k(\cdot), \kappa_k(\cdot, \cdot)) \quad \text{for} \quad k = 1, \ldots, K$$

$$h_n^{(r)}(\cdot) = \sum_{k=1}^{K} c_{n,k} x_k^{(r)}(\cdot) + d_n \quad \text{for} \quad n = 1, \ldots, N \quad (1)$$

$$y_n^{(r)}(t_i) \sim p(y_n^{(r)}(t_i)|h_n^{(r)}(t_i)) \quad \text{for} \quad i = 1, \ldots, T$$

The $c_{n,k}$ are weights for each latent and neuron that define a subspace mapping from low-dimensional latent space to high-dimensional neural space and $d_n$ is a constant offset.

The widespread use of GPFA may be restricted by its poor scaling with time: Since time is evaluated on an evenly spaced grid with $T$ points, GPFA requires building and inverting a $T \times T$ covariance matrix, leading to $\mathcal{O}(T^3)$ complexity. The intractability of performing exact inference in GPFA

models with non-conjugate likelihood adds further complexity on top of this [23, 24]. In the next section, we will outline how to improve the scalability of GPFA irrespective of the choice of observation model via the use of inducing points.

# 3  Sparse Variational Gaussian Process Factor Analysis (svGPFA)

The framework of sparse variational GP approximations [25] has helped to overcome difficulties associated with the scalability of GP methods to large sample sizes. It has since been applied to diverse problems in GP inference, contributing to improvements of the scalability of GP methods to large datasets and complex, potentially non-conjugate, applications [26–30].

A sparse variational extension of the GPFA model can be obtained by extending the work on additive signal decomposition in [26]. The main idea is to augment the model in (1) by introducing inducing points $\boldsymbol{u}_k$ for each latent process $k = 1, \ldots, K$. The inducing points $\boldsymbol{u}_k$ represent function evaluations of the $k$th latent GP at $M_k$ input locations $\boldsymbol{z}_k$. A joint prior over the process $x_k(\cdot)$ and its inducing points can hence be written as

$$
\begin{aligned}
p(\boldsymbol{u}_k|\boldsymbol{z}_k) &= \mathcal{N}\left(\boldsymbol{u}_k|\boldsymbol{0}, \mathsf{K}_{zz}^{(k)}\right) \\
p(x_k(\cdot)|\boldsymbol{u}_k) &= \mathcal{GP}(\tilde{\mu}_k(\cdot), \tilde{\kappa}_k(\cdot, \cdot))
\end{aligned}
\tag{2}
$$

where the GP prior over $x_k(\cdot)$ is now conditioned on the inducing points with conditional mean and covariance function

$$
\begin{aligned}
\tilde{\mu}_k(t) &= \boldsymbol{\kappa}_k(t, \boldsymbol{z}) \mathsf{K}_{zz}^{(k)-1} \boldsymbol{u}_k \\
\tilde{\kappa}_k(t, t') &= \kappa_k(t, t') - \boldsymbol{\kappa}_k(t, \boldsymbol{z}_k) \mathsf{K}_{zz}^{(k)-1} \boldsymbol{\kappa}_k(\boldsymbol{z}_k, t')
\end{aligned}
\tag{3}
$$

Here, $\boldsymbol{\kappa}_k(\cdot, \boldsymbol{z})$ is a vector valued function taking a single input argument and consisting of evaluations of the covariance function $\kappa_k(\cdot, \cdot)$ at the input and inducing point locations $\boldsymbol{z}_k$; $\mathsf{K}_{zz}^{(k)}$ is the Gram matrix of $\kappa_k(\cdot, \cdot)$ evaluated at the inducing point locations.

We follow [26] in choosing a factorised variational approximation for posterior inference of the form $q(\boldsymbol{u}_{1:K}, x_{1:K}) = \prod_{k=1}^{K} p(x_k|\boldsymbol{u}_k)q(\boldsymbol{u}_k)$, with Gaussian $q(\boldsymbol{u}_k) = \mathcal{N}(\boldsymbol{u}_k|\boldsymbol{m}_k, S_k)$. This choice of posterior approximation makes it possible to derive a variational lower bound to the marginal log-likelihood over the observed data $\mathcal{Y} = \{\boldsymbol{y}_1^{(r)}, \ldots, \boldsymbol{y}_N^{(r)}\}_{r=1}^{R}$ of the form

$$
\log p(\mathcal{Y}) \geq \sum_{r=1}^{R} \sum_{n=1}^{N} \mathbb{E}_{q(h_n^{(r)})}\left[\log p(\boldsymbol{y}_n^{(r)}|h_n^{(r)})\right] - \sum_{r=1}^{R} \sum_{k=1}^{K} \mathrm{KL}\left[q(\boldsymbol{u}_k^{(r)})\|p(\boldsymbol{u}_k^{(r)})\right] \stackrel{def}{=} \mathcal{F} \tag{4}
$$

where $q(h_n^{(r)})$ is the variational distribution over the affine transformation of the latents for the $n$th neuron obtained from $q(x) = \int p(x|\boldsymbol{u})q(\boldsymbol{u})d\boldsymbol{u}$. $q(h_n^{(r)})$ is a GP with additive structure. Its mean function $\nu_n^{(r)}(t)$ and covariance function $\sigma_n^{(r)}(t, t')$ are given by

$$
\begin{aligned}
\nu_n^{(r)}(t) &= \sum_k c_{n,k}\, \boldsymbol{\kappa}_k(\,t\,, \boldsymbol{z}_k)\, \mathsf{K}_{zz}^{(k)-1}\, \boldsymbol{m}_k^{(r)} + d_n \\
\sigma_n^{(r)}(t, t') &= \sum_k c_{n,k}^2 \left(\kappa_k(t, t') + \boldsymbol{\kappa}_k(t, \boldsymbol{z}_k)\left(\mathsf{K}_{zz}^{(k)-1} S_k^{(r)} \mathsf{K}_{zz}^{(k)-1} - \mathsf{K}_{zz}^{(k)-1}\right)\boldsymbol{\kappa}_k(\boldsymbol{z}_k, t')\right)
\end{aligned}
\tag{5}
$$

The cost of evaluating this bound on the likelihood now scales linearly in the number of time points $T$, with cubic scaling only in the number of inducing points. Maximising the lower bound $\mathcal{F}$ in (4) allows for variational learning of the parameters in $q(\boldsymbol{u}_k)$, the kernel hyperparmeters, the inducing point locations, and the model parameters describing the affine transformation from latents to $h_n$s.

## 3.1  A continuous-time point-process observation model

The form of the variational lower bound in (4) makes it clear that including different observation models only requires taking a Gaussian expectation of the respective log-likelihood. Importantly, the inference approach is essentially decoupled from the locations of the observed data. This crucial

consequence of the inducing point approach makes it possible to move away from gridded, binned data and fully exploit the power of GPs in continuous-time.

Previous work has used sparse variational GP approximations to infer the intensity of a univariate GP modulated Poisson process [27]. Here, we extend this to the multivariate case by combining the svGPFA model with a point-process likelihood. To do this, we relate the affine transformation of the latent processes for the $n$th neuron, $h_n(\cdot)$, to the non-negative intensity function of a point-process, $\lambda_n(\cdot)$, via a static non-linearity $g : \mathbb{R} \to \mathbb{R}^+$. Thus, the spike times of neuron $n$ on trial $r$, $\boldsymbol{t}_n^{(r)} = \{t_1, \ldots t_{\Phi(n,r)}\}$, are modelled as

$$p(\boldsymbol{t}_n^{(r)}|\lambda_n^{(r)}) = \exp\left(-\int_0^{\mathcal{T}_r} dt\, \lambda_n^{(r)}(t)\right) \prod_{i=1}^{\Phi(n,r)} \lambda_n^{(r)}(t_i) \tag{6}$$

Where $\lambda_n(t) = g(h_n(t))$, $\mathcal{T}_r$ is the duration of the $r$th trial, and $\Phi(n,r)$ is the total spike-count of neuron $n$ on trial $r$. The expected log-likelihood term in (4) can be evaluated as

$$\mathbb{E}_{q(h_n^{(r)})}\left[\log p(\boldsymbol{t}_n^{(r)}|h_n^{(r)})\right] = -\int_0^{\mathcal{T}_r} \mathbb{E}_{q(h_n^{(r)})}\left[g(h_n^{(r)}(t))\right] dt + \sum_{i=1}^{\Phi(n,r)} \mathbb{E}_{q(h_n^{(r)})}\left[\log g(h_n^{(r)}(t_i))\right] \tag{7}$$

The resulting expected log-likelihood in (7) still contains an integral over the expected rate function of the neuron, which cannot be computed analytically. However, the integral is one-dimensional and can be computed numerically using efficient quadrature rules [31, 32].

The svGPFA model with point-process likelihood already fully addresses the two major limitations of the classic GPFA approach outlined in section 2: Firstly, it improves the scalability of the algorithm via the use of inducing points, scaling cubically only in the number of inducing points per latent and linearly in the total number of spiking events. Secondly, our approach appropriately models neural spike trains as observations of a point-process. The model also provides the basis for further extensions addressing temporal alignment across trials, which will be the focus of the following section.

## 4 Temporal alignment and latent factor inference using Gaussian processes

The svGPFA model we have developed in section 3 aims to extract different latent trajectories on each trial. It does not explicitly model any structure that is shared across trials, and each trial's variable time-course is simply captured via inter-trial differences in latents. In this section, we will extend our basic svGPFA model in order to disentangle inter-trial variations in time-course from variations in the latent trajectories themselves. To achieve this, we explicitly model latent structure that is shared across trials or subsets of trials, as well as structure that is specific to each trial, and make use of a nested GP architecture with time-warping functions. In this way, shared, neurally-defined latent structure provides an anchor for the temporal alignment across trials. We extend the previous inducing point approach in order to arrive at a sparse variational inference algorithm in this setting.

### 4.1 A generative model for population spike times with grouped trial structure and variable time courses

We introduce latent processes that are shared across all trials, across subsets of trials that share the same experimental condition, or that are specific to each individual trial. We model each of these as draws from a GP prior with $K$ latent processes, $L$ experimental conditions, and $R$ trials:

$$\begin{aligned}
\text{shared:} \qquad & \alpha_k(\cdot) \sim \mathcal{GP}(\mu_k^\alpha(\cdot), \kappa_k^\alpha(\cdot, \cdot)) \quad \text{for} \quad k = 1, \ldots, K \\
\text{condition-specific:} \qquad & \beta_k^{(\ell)}(\cdot) \sim \mathcal{GP}(\mu_k^\beta(\cdot), \kappa_k^\beta(\cdot, \cdot)) \quad \text{for} \quad \ell = 1, \ldots, L \\
\text{trial-specific:} \qquad & \gamma_k^{(r)}(\cdot) \sim \mathcal{GP}(\mu_k^\gamma(\cdot), \kappa_k^\gamma(\cdot, \cdot)) \quad \text{for} \quad r = 1, \ldots, R
\end{aligned} \tag{8}$$

Allowing each of these latents to evolve in potentially separate subspaces, we define the linear mapping from low-dimensional latent, to high-dimensional neural space to be of the form

$$h_n^{(r)}(\cdot) = \sum_{k=1}^K \left( c_{n,k}^\alpha \alpha_k(\cdot) + c_{n,k}^\beta \beta_k^{\ell(r)}(\cdot) + c_{n,k}^\gamma \gamma_k^{(r)}(\cdot) \right) + d_n \tag{9}$$

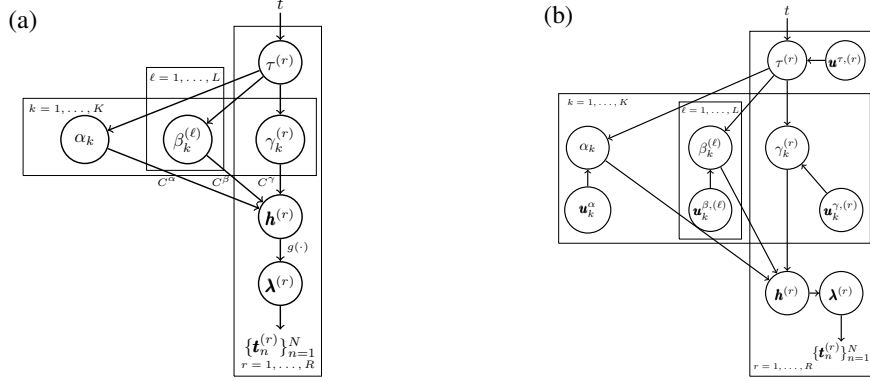

Figure 1: (a) generative model architecture. (b) the augmented model with inducing points.

where $\ell(r)$ is the condition of trial $r$. When these latents are evaluated on a canonical time-axis all trials evolve according to the same time-course. To incorporate inter-trial variability in time-course into our model, we include a second GP that acts to warp time differently on each trial:

$$\text{time-warping:} \qquad \tau^{(r)}(\cdot) \sim \mathcal{GP}(\mu^\tau(\cdot), \kappa^\tau(\cdot, \cdot)) \quad \text{for} \quad r = 1, \dots, R \qquad (10)$$

As before, we can relate $h_n(\cdot)$ to the neuron's firing rate via a static non-linearity $g(\cdot)$, and use a point-process observation model for spike times. Thus, the firing rate of neuron $n$ on trial $r$ is described as $\lambda_n^{(r)}(t) = g(h_n^{(r)}(\tau^{(r)}(t)))$. This generative model is summarised in Figure 1(a).

## 4.2 Sparse variational inference using inducing points

The generative model introduced in section 4.1 is a two-layer GP, where one layer has additive structure. Previous work in the GP literature has successfully applied inducing point approaches for variational inference in nested GP architectures [33, 34] (albeit without explicit additive decompositions within layers), and we can take a similar approach here.

We introduce sets of inducing points for each of the latent processes $\alpha_k(\cdot)$, $\beta_k^{(\ell)}(\cdot)$, $\gamma_k^{(r)}(\cdot)$, and $\tau^{(r)}(\cdot)$, which we will denote as $\boldsymbol{u}_k^\alpha$, $\boldsymbol{u}_k^{\beta,(\ell)}$, $\boldsymbol{u}_k^{\gamma,(r)}$, and $\boldsymbol{u}^{\tau,(r)}$, respectively. The augmented model using inducing points is summarised in Figure 1(b). For each of the latent processes $\zeta = \{\alpha, \beta, \gamma\}$ we choose a factorised approximating distribution of the form $q(\{\zeta_k, \boldsymbol{u}_k^\zeta\}_{k=1}^K | \tau) = \prod_{k=1}^K p(\zeta_k | \boldsymbol{u}_k^\zeta, \tau) q(\boldsymbol{u}_k^\zeta)$ with $q(\boldsymbol{u}_k^\zeta) = \mathcal{N}(\boldsymbol{u}_k^\zeta | \boldsymbol{m}_k^\zeta, S_k^\zeta)$. For the time-warping process, we choose $q(\tau, \boldsymbol{u}^\tau) = p(\tau | \boldsymbol{u}^\tau) q(\boldsymbol{u}^\tau)$ with $q(\boldsymbol{u}^\tau) = \mathcal{N}(\boldsymbol{u}^\tau | \boldsymbol{m}^\tau, S^\tau)$.

Using this approximation, the variational lower bound to the marginal log-likelihood becomes

$$
\begin{aligned}
\mathcal{F} = &\sum_{r,n} \mathbb{E}_{q(h_n^{(r)})} \left[ \log p(\boldsymbol{y}_n^{(r)} | h_n^{(r)}) \right] - \sum_r \text{KL} \left[ q(\boldsymbol{u}^{\tau,(r)}) \| p(\boldsymbol{u}^{\tau,(r)}) \right] \\
&- \sum_k \text{KL} \left[ q(\boldsymbol{u}_k^\alpha) \| p(\boldsymbol{u}_k^\alpha) \right] - \sum_{\ell,k} \text{KL} \left[ q(\boldsymbol{u}_k^{\beta,(\ell)}) \| p(\boldsymbol{u}_k^{\beta,(\ell)}) \right] - \sum_{r,k} \text{KL} \left[ q(\boldsymbol{u}_k^{\gamma,(r)}) \| p(\boldsymbol{u}_k^{\gamma,(r)}) \right]
\end{aligned}
$$

$$(11)$$

The mean and covariance function of the variational GP $q(h_n^{(r)})$ are given by

$$\nu_n^{(r)}(t) = \sum_{\zeta,k} c_{n,k}^\zeta \, \Psi_{k,1}^{\zeta,(r)}(t, \boldsymbol{z}_k^\zeta) \, \mathsf{K}_{zz}^{\zeta,(k)^{-1}} \, \boldsymbol{m}_k^{\zeta,(r)} + d_n$$

$$\sigma_n^{(r)}(t,t) = \sum_{\zeta,k} c_{n,k}^{\zeta\,2} \left( \Psi_{k,0}^{\zeta,(r)}(t) + \text{Tr} \left[ \left( \mathsf{K}_{zz}^{\zeta,(k)^{-1}} S_k^{\zeta,(r)} \mathsf{K}_{zz}^{\zeta,(k)^{-1}} - \mathsf{K}_{zz}^{\zeta,(k)^{-1}} \right) \Psi_{k,2}^{\zeta,(r)}(t, \boldsymbol{z}_k^\zeta) \right] \right)$$

$$(12)$$

The $\Psi_{k,i}^{\zeta,(r)}(t,\boldsymbol{z}_k^\zeta)$ are the $\Psi$-statistics [33, 35] of the kernel covariance functions

$$\Psi_{k,0}^{\zeta,(r)}(t) = \mathbb{E}_{q(\tau^{(r)})}\left[\boldsymbol{\kappa}_k^\zeta(\,\tau^{(r)}(t)\,,\,\tau^{(r)}(t))\right]$$

$$\Psi_{k,1}^{\zeta,(r)}(t,\boldsymbol{z}_k^\zeta) = \mathbb{E}_{q(\tau^{(r)})}\left[\boldsymbol{\kappa}_k^\zeta(\,\tau^{(r)}(t)\,,\boldsymbol{z}_k^\zeta)\right] \tag{13}$$

$$\Psi_{k,2}^{\zeta,(r)}(t,\boldsymbol{z}_k^\zeta) = \mathbb{E}_{q(\tau^{(r)})}\left[\boldsymbol{\kappa}_k^\zeta(\boldsymbol{z}_k^\zeta,\tau^{(r)}(t))\boldsymbol{\kappa}_k^\zeta(\tau^{(r)}(t),\boldsymbol{z}_k^\zeta)\right]$$

The $\Psi$-statistics can be evaluated analytically for common kernel choices such as the linear, exponentiated quadratic, or cosine kernel. For other kernel choices they can be computed using e.g. Gaussian quadrature. Finally, the variational distribution over the time-warping functions $q(\tau^{(r)}) = \int p(\tau^{(r)}|\boldsymbol{u}^{\tau,(r)})q(\boldsymbol{u}^{(\tau,(r)})d\boldsymbol{u}^{(\tau,(r)}$ is also a GP with mean and covariance function

$$\nu^{\tau,(r)}(t) = \boldsymbol{\kappa}^\tau(\,t\,,\boldsymbol{z}^\tau)\,\mathsf{K}_{zz}^{\tau}{}^{-1}\,\boldsymbol{m}^{\tau,(r)}$$

$$\sigma^{\tau,(r)}(t,t') = \kappa^\tau(t,t') + \boldsymbol{\kappa}^\tau(t,\boldsymbol{z}^\tau)\left(\mathsf{K}_{zz}^{\tau}{}^{-1}S^{\tau,(r)}\mathsf{K}_{zz}^{\tau}{}^{-1} - \mathsf{K}_{zz}^{\tau}{}^{-1}\right)\boldsymbol{\kappa}^\tau(\boldsymbol{z}^\tau,t'). \tag{14}$$

The variational lower bound in (11) can thus be evaluated tractably and optimised with respect to all parameters in the model. While the decomposition into shared, condition-specific and trial-specific latents and the addition of the time-warping layer increases the total number of inducing points that require optimisation, the impact on the time-complexity of the algorithm is minimal: it remains linear in the total number of spikes across trials and neurons, and only cubic in the number of inducing points *per individual latent process*.

## 5 Results

### 5.1 Synthetic data

We first generate synthetic data for 100 neurons on 15 trials that are each one second in duration. The neural activity is driven by one shared and one condition-specific latent process. We omit the trial-specific latent process here, such that inter-trial variability is solely due to the variable time-course of each trial and independent Poisson variability in the spiking of each neuron. The generative procedure is illustrated in Figure 2 and aims to provide a toy simulation of decision-making, where trial-to-trial differences in the decision-making process are modelled via the time-warping.

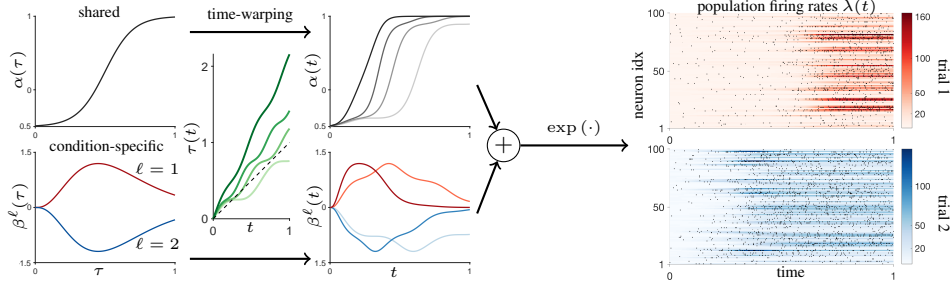

Figure 2: Synthetic data example: the shared latent reflects a gating signal; conditions are preferred and anti-preferred for each neuron. Inter-trial differences in the decision-making process are modelled via time warping. Warped latents are shown for two example trials per condition. The warped latents are mixed linearly and passed through an exponential non-linearity to generate firing rates, which drive a Poisson process. Example spike rasters are superimposed over the population firing rates for two example trials of different conditions.

We first investigate how our sparse variational inference approach compares to the Poisson GPFA approach proposed in [23] (vLGP), using published code[1] which includes some further numerical approximations for speed. Figure 3(a) shows that the svGPFA methods achieve an improved approximation at lower or comparable runtime cost. We next fit the time-warped method (tw-pp-svGPFA)

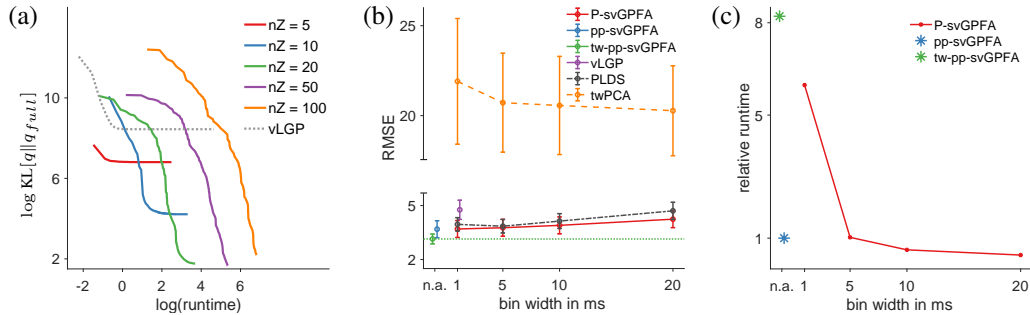

Figure 3: (a) runtime in seconds vs. approximation error of inference approach (under generative parameters and matched kernel hyperparameters) of pp-svGPFA (solid) with different numbers of inducing points (nZ) and vLGP (dashed). Approximation error is computed as the mean KL-divergence between the respective marginal distributions $q(h_n(t))$ and those of a full Gaussian variational approximation over latent processes. (b) comparison of RMSE of inferred firing rates across different methods and bin widths. K = 2 for all methods. (c) relative runtime comparisons across svGPFA methods.

with 5 inducing points for the shared and condition-specific latents, and 8 inducing points for the time-warping latent. We use Gauss-Legendre quadrature with 80 abscissas for evaluating the log-normaliser term in (7). We compare tw-pp-svGPFA with time-warped PCA [10] (twPCA)[2], vLGP, a latent linear dynamical systems model with Poisson observations (PLDS) [5], and our point-process (pp-) and Poisson (P-) svGPFA model without time-warping, but with all other settings chosen equivalently. For the discrete-time methods, we bin the spike trains at different resolutions before applying either method to the data.

Figure 3(b) shows the RMSE in inferred firing rates of each method. The svGPFA variants achieve the best posterior estimate of firing rates throughout. The svGPFA models without time-warping and vLGP have to fit the temporal variability via inter-trial differences in the latent processes. The PLDS model captures the discrete-time evolution of the latent using a linear dynamical operator and an additive noise process, often referred to as the innovations. It is this innovations process that allows for inter-trial variability in this case. Thus, neither of these competing approaches offers a dissociation of the variations in time-course from other sources of inter-trial variability. In the case of the PLDS, discrepancies between the inferred latent path, and a canonical path as predicted via the learnt dynamical operator could in principle be used to estimate an alignment across trials. However, innovations noise is typically modelled via a stationary parameter which restricts the capacity of such an approach to capture more complex variations in timing. The twPCA approach does not assume a noise model for the observations, which limits its ability to recover firing rates and time-warped structure underlying variable spike trains. This is especially true as spiking is relatively sparse in our example. Figure 3(c) compares the relative runtime between the different svGPFA models.

## 5.2 Variable-delay centre-out reaching task

We apply tw-pp-svGPFA to simultaneously recorded spike trains of 105 neurons in dorsal premotor and primary motor cortex of a macaque monkey (*Macaca mulatta*) during a variable-delay centre-out reaching task [2].

In this task, the animal is presented with one of multiple possible target locations, but has to wait for a variable duration before it receives the instruction to perform an arm-reach to the target. Beyond these designed sources of inter-trial variability, there are also variations in reaction time and movement kinematics, as well as extra variability in neural firing.

We fit our model to repeated trials from 5 different target directions (12 trials per condition). We use 20 inducing points per latent, 10 inducing points for the time-warping functions, and 100 Gauss-Legendre quadrature abscissas to evaluate the log-normaliser term in (7).

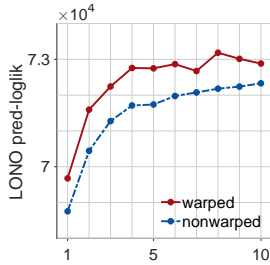

Figure 4: Leave-one-neuron-out (LONO) cross-validation across latent dimensionality on 15 held-out trials (3 per condition). Results are shown for models with an exponential static non-linearity, with and without the time-warping layer. The time-warped models achieve a higher predictive log-likelihood than their non-warped analogues.

We perform model comparison across latent dimensionality by leave-one-neuron-out cross-validation on held-out data [2]. To demonstrate the benefit of time-warping, we compare the time-warped models to their svGPFA analogues without a time-warping layer. Figure 4 shows that time-warping results in a substantial increase in predictive log-likelihood on held-out data.

Figure 5(a) shows that the inferred time-warping functions automatically align the latent trajectories to the onset of movement (MO). The model did not have access to this information and the alignment is purely based on structure present in the neural population activity. Furthermore, the time passed in warped time since the go-cue (GC) is highly predictive of reaction times, as is shown in Figure 5(b). We achieve an $R^2$ value of 0.90, which is substantially higher than previously reported values on the same dataset ([36] report $R^2 = 0.52$ using a switching LDS model). Figure 5(c) shows a reduction in the standard deviation of the times when the animal's absolute hand position crosses a threshold value. This reduction is not simply due to an overall compression of time, since the slope of the time-warping functions during the movement period was 1.0572 on average across trials. This demonstrates that the inferred time-warping provides an improved alignment of the behaviourally measured arm reach compared to MO or GC alignment, which are common ways of aligning data in reaching experiments [37].

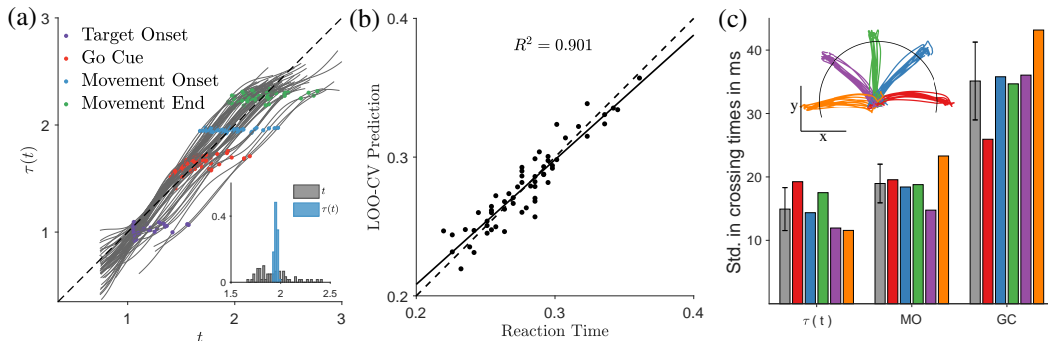

Figure 5: (a) Inferred time-warping functions and behavioural events. $\tau(t)$ automatically aligns trials to MO. Inset histogram compares the distribution of MO times in measured, and in warped time. (b) leave-one-out cross-validated predictions of the reaction times, computed by determining a threshold value for MO in warped time from all but one trials, and taking the time it takes to pass this threshold from the GC on the held-out trial. (c) within-condition standard deviation (std.) in times when the absolute hand position crosses a threshold under different methods of alignment. Inset illustrates target directions and location of the threshold (85% of average reach distance). Grey bars show average std. across target directions.

## 6 Conclusion

In this work, we have introduced a scalable sparse variational extension of GPFA, allowing one to extract latent structure directly from the unbinned spike-time observations of simultaneously recorded neurons. We further extended GPFA using an explicit separation of shared variability into condition-dependent and trial-specific components within a nested GP architecture. This allowed us to separate trial-to-trial variations in trajectories from variations in time-course. We arrived at a scalable algorithm for posterior inference using inducing points.

Using synthetic data, we showed that our svGPFA methods can more accurately recover firing rate estimates than related methods. Using neural data, we demonstrated that our time-warped method infers a behaviourally meaningful alignment in a completely unsupervised fashion using only information present in the neural population spike times. We showed that the inferred time-warping is behaviourally meaningful, highly predictive of reaction times, and provides an improved temporal alignment compared to manually aligning to behaviourally defined time-points.

While we have focused on temporal variability in this work, we note that the inference scheme using inducing points is more generally applicable for a variety of other nested GP models of interest in neuroscience, such as the GP-LVM [35, 38]. Similarly, further extensions of our model with time-warping to deeper GP hierarchies – for instance using a non-linear mapping from latents to observations as in the GP-LVM – can be incorporated straightforwardly.

The problem of latent input alignment in the context of the GP-LVM has also recently been considered in [39], where maximum a posteriori inference in a Gaussian model is used to recover a single true underlying input sequence from multiple observations that are generated from different time-warping functions. The sparse-variational approach we have presented here could hence be used to generalise this approach to account for further differences across sequences, and extend it to non-Gaussian settings.

Furthermore, our approach can also be applied to multiple sets of observed data with potentially different likelihoods in a canonical correlations analysis extension of GPFA. This could be useful for inferring latent structure from multiple recorded signals. For instance, one could combine the analysis of LFP or calcium imaging data with that of spike trains. Thus, with slight modifications to the model architecture, an analogous inference approach to the one presented here is applicable in a variety of contexts, including manifold learning, combinations of modalities, and beyond.

### Acknowledgements

We would like to thank Vincent Adam for early contributions to this project, Gergö Bohner for helpful discussions, and the Shenoy laboratory at Stanford University for sharing the centre-out reaching dataset. This work was funded by the Simons Foundation (SCGB 323228, 543039; MS) and the Gatsby Charitable Foundation.

## Footnotes

[1] `https://github.com/catniplab/vlgp`

[2]https://github.com/ganguli-lab/twpca

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
