[Supplementary Material]

# Supplementary Material:
# Temporal alignment and latent Gaussian process factor inference in population spike trains

**Lea Duncker & Maneesh Sahani**
Gatsby Computational Neuroscience Unit
University College London
London, W1T 4JG
{duncker,maneesh}@gatsby.ucl.ac.uk

## 1  Derivation of the svGPFA Variational Lower Bound

In order to arrive at a scalable variational inference algorithm, we make use of a sparse GP approximation. We introduce a set of inducing points $U = [\boldsymbol{u}_1, \ldots, \boldsymbol{u}_K]$ for each latent process, which are each evaluated on a set of inducing point locations $Z = [\boldsymbol{z}_1, \ldots, \boldsymbol{z}_K]$.

The joint-data likelihood of the full model, now including the inducing points, is hence

$$p(\mathcal{Y}, \boldsymbol{x}(\cdot), U) = p(\mathcal{Y}|\boldsymbol{x}(\cdot)) \prod_{k=1}^{K} p(x_k(\cdot)|\boldsymbol{u}_k) p(\boldsymbol{u}_k|\boldsymbol{z}_k) \tag{1}$$

where we have omitted explicit conditioning on model parameters to avoid cluttered notation. A variational lower bound to the log-likelihood can be obtained by applying Jensen's inequality:

$$\log p(\mathcal{Y}) \geq \iint q(\{\boldsymbol{u}_k\}_{k=1}^{K}, \boldsymbol{x}(\cdot)) \log \left( \frac{p(\mathcal{Y}|\boldsymbol{x}(\cdot)) \prod_{k=1}^{K} p(x_k(\cdot)|\boldsymbol{u}_k) p(\boldsymbol{u}_k|\boldsymbol{z}_k)}{q(\{\boldsymbol{u}_k\}_{k=1}^{K}, \boldsymbol{x}(\cdot))} \right) d\boldsymbol{u}_1 \ldots d\boldsymbol{u}_K d\boldsymbol{x} \tag{2}$$

We choose a factorised approximating distribution of the form

$$q(\{\boldsymbol{u}_k\}_{k=1}^{K}, \boldsymbol{x}(\cdot)) = \prod_{k=1}^{K} q(\boldsymbol{u}_k, x_k(\cdot)) = \prod_{k=1}^{K} p(x_k(\cdot)|\boldsymbol{u}_k) q(\boldsymbol{u}_k) \tag{3}$$

and choose $q(\boldsymbol{u}_k) = \mathcal{N}(\boldsymbol{m}_k, S_k)$ to be multivariate Gaussian. This choice of approximating distribution allows one to write the lower bound as

$$\log p(\mathcal{Y}) \geq \iint \prod_{k=1}^{K} q(\boldsymbol{u}_k) p(x_k(\cdot)|\boldsymbol{u}_k) \log \left( p(\mathcal{Y}|\boldsymbol{x}(\cdot)) \right) d\boldsymbol{u}_k d\boldsymbol{x} - \sum_{k=1}^{K} \text{KL}\left[ q(\boldsymbol{u}_k) \| p(\boldsymbol{u}_k) \right] \tag{4}$$

The second term is the Kullback-Leibler divergence between two Gaussian distributions, which can be evaluated analytically. In order to manipulate the first term, let $h_n(\cdot) = \sum_{k=1}^{K} c_{n,k} x_k(\cdot) + d_n$ denote the affine transformation of latent GPs for the $n$-th neuron. We can obtain a marginal variational distribution over $h_n(\cdot)$ as a GP with additive structure

$$q(h_n(\cdot)) = \int \prod_{k=1}^{K} q(\boldsymbol{u}_k) p(h_n(\cdot)|\{\boldsymbol{u}_k\}) d\boldsymbol{u}_k = \mathcal{GP}(\nu_n(\cdot), \sigma_n(\cdot, \cdot)) \tag{5}$$

where

$$\nu_n(t) = \sum_{k=1}^{K} c_{n,k}\, \boldsymbol{\kappa}_k(\,t\,,\boldsymbol{z}_k)\, \mathsf{K}_{zz}^{(k)^{-1}}\, \boldsymbol{m}_k + d_n \tag{6}$$

$$\sigma_n(t,t') = \sum_{k=1}^{K} c_{n,k}^2 \left( \kappa_k(t,t') + \boldsymbol{\kappa}_k(t,\boldsymbol{z}_k)\left( \mathsf{K}_{zz}^{(k)^{-1}} S_k \mathsf{K}_{zz}^{(k)^{-1}} - \mathsf{K}_{zz}^{(k)^{-1}} \right)\boldsymbol{\kappa}_k(\boldsymbol{z}_k,t') \right) \tag{7}$$

$\boldsymbol{\kappa}_k(\,\cdot\,,\boldsymbol{z}_k)$ is a vector valued function taking a single time-point as an input argument and consisting of evaluations of the kernel $\kappa_k(\cdot,\cdot)$ at the inducing point locations. $\mathsf{K}_{zz}^{(k)}$ is the kernel Gram matrix of the $k$th process evaluated at the respective inducing point locations.

To obtain the final expression for the variational lower bound, we can simply rewrite the expression in equation (**??**):

$$\log p(\mathcal{Y}) \geq \mathbb{E}_{q(h_n)}\left[\log p(\mathcal{Y}|h_n(\cdot))\right] - \sum_{k=1}^{K} \mathrm{KL}\big[q(\boldsymbol{u}_k)\|p(\boldsymbol{u}_k)\big] \tag{8}$$

## 2 Use of a point-process likelihood

Using a point-process likelihood in the GPFA model amounts to evaluating the expected log-likelihood in the first term in (**??**):

$$\mathbb{E}_{q(h_n^{(r)})}\left[\log p(\boldsymbol{t}_n^{(r)}|h_n^{(r)})\right] = -\mathbb{E}_{q(h_n^{(r)})}\left[\int_{\mathcal{T}} g(h_n^{(r)}(t))dt\right] + \sum_{i_n=1}^{\Phi(n,r)} \mathbb{E}_{q(h_n^{(r)})}\left[\log g(h_n^{(r)}(t_i))\right] \tag{9}$$

We can apply Fubini's theorem to switch the order of integration in the first term:

$$\mathbb{E}_{q(h_n^{(r)})}\left[\log p(\boldsymbol{t}_n^{(r)}|h_n^{(r)})\right] = -\int_{\mathcal{T}} \mathbb{E}_{q(h_n^{(r)})}\left[g(h_n^{(r)}(t))\right]dt + \sum_{i_n=1}^{\Phi(n,r)} \mathbb{E}_{q(h_n^{(r)})}\left[\log g(h_n^{(r)}(t_i))\right] \tag{10}$$

Which gives the final form of the expected log-likelihood. Depending on the choice of non-linearity $g(\cdot)$, the expectation terms can either be evaluated analytically, or efficiently using Gauss-Hermite quadrature. The first term in (**??**) involves one-dimensional integrals, which can be computed using efficient numerical approximations such as Gauss-Legendre quadrature.

## 3 Condition-grouped model with time-warping

The full approximating distribution across trials is chosen to be of the form

$$q(\{\{\{\zeta_k^{(r)}, \boldsymbol{u}_k^{\zeta,(r)}\}_{k=1}^{K}\}_\zeta, \tau^{(r)}, \boldsymbol{u}^{\tau,(r)}\}_{r=1}^{R})$$

$$= \prod_{r=1}^{R} \left( \prod_{\zeta=\{\alpha,\beta,\gamma\}} \prod_{k=1}^{K} p(\zeta_k^{(r)}|\boldsymbol{u}_k^{\zeta,(r)}, \tau^{(r)})q(\boldsymbol{u}_k^{\zeta,(r)}) \right) p(\tau^{(r)}|\boldsymbol{u}^{\tau,(r)})q(\boldsymbol{u}^{\tau,(r)}) \tag{11}$$

Under this approximation, the variational lower bound to the log-likelihood becomes

$$\log p(\mathcal{Y}) \geq \sum_{r=1}^{R}\sum_{n=1}^{N} \mathbb{E}_{q(h_n^{(r)})}\left[\log p(\boldsymbol{y}_n^{(r)}|h_n^{(r)})\right] - \sum_{r=1}^{R} \mathrm{KL}\left[q(\boldsymbol{u}^{\tau,(r)})\|p(\boldsymbol{u}^{\tau,(r)})\right]$$

$$- \sum_{k=1}^{K} \mathrm{KL}\left[q(\boldsymbol{u}_k^{\alpha})\|p(\boldsymbol{u}_k^{\alpha})\right] - \sum_{\ell=1}^{L}\sum_{k=1}^{K} \mathrm{KL}\left[q(\boldsymbol{u}_k^{\beta,(\ell)})\|p(\boldsymbol{u}_k^{\beta,(\ell)})\right] \tag{12}$$

$$- \sum_{r=1}^{R}\sum_{k=1}^{K} \mathrm{KL}\left[q(\boldsymbol{u}_k^{\gamma,(r)})\|p(\boldsymbol{u}_k^{\gamma,(r)})\right]$$

Where

$$q(h_n^{(r)}(t))$$
$$= \int d\boldsymbol{u}_k^{\alpha} d\boldsymbol{u}_k^{\beta,\ell(r)} d\boldsymbol{u}_k^{\gamma,(r)} d\boldsymbol{u}^{\tau,(r)} d\tau^{(r)}$$

$$\prod_{k=1}^{K} q(\boldsymbol{u}_k^{\alpha}) q(\boldsymbol{u}_k^{\beta,\ell(r)}) q(\boldsymbol{u}_k^{\gamma,(r)}) p(h_n^{(r)}(t)|\{\boldsymbol{u}_k^{\zeta,(r)}\}_{k,\zeta}, \tau^{(r)}) p(\tau^{(r)}|\boldsymbol{u}^{\tau,(r)}) q(\boldsymbol{u}^{\tau,(r)}) \tag{13}$$

Letting

$$q(\tau^{(r)}) = \int d\boldsymbol{u}^{\tau,(r)} p(\tau^{(r)}|\boldsymbol{u}^{\tau,(r)}) q(\boldsymbol{u}^{\tau,(r)}) \tag{14}$$

We can marginalise out the inducing points and evaluate $q(h_n^{(r)}(t))$ as an additive Gaussian Process with mean and covariance function:

$$\nu_n^{(r)}(t) = \sum_{\zeta,k} c_{n,k}^{\zeta} \ \Psi_{k,1}^{\zeta,(r)}(t, \boldsymbol{z}_k^{\zeta}) \ \mathsf{K}_{zz}^{\zeta,(k)^{-1}} \ \boldsymbol{m}_k^{\zeta,(r)} + d_n$$

$$\sigma_n^{(r)}(t,t) = \sum_{\zeta,k} c_{n,k}^{\zeta}{}^2 \left( \Psi_{k,0}^{\zeta,(r)}(t) + \mathsf{Tr}\left[ \left( \mathsf{K}_{zz}^{\zeta,(k)^{-1}} S_k^{\zeta,(r)} \mathsf{K}_{zz}^{\zeta,(k)^{-1}} - \mathsf{K}_{zz}^{\zeta,(k)^{-1}} \right) \Psi_{k,2}^{\zeta,(r)}(t, \boldsymbol{z}_k^{\zeta}) \right] \right)$$

$$\tag{15}$$

where

$$\Psi_{k,0}^{\zeta,(r)}(t) = \mathbb{E}_{q(\tau^{(r)})} \left[ \boldsymbol{\kappa}_k^{\zeta}( \tau^{(r)}(t) , \tau^{(r)}(t)) \right]$$

$$\Psi_{k,1}^{\zeta,(r)}(t, \boldsymbol{z}_k^{\zeta}) = \mathbb{E}_{q(\tau^{(r)})} \left[ \boldsymbol{\kappa}_k^{\zeta}( \tau^{(r)}(t) , \boldsymbol{z}_k^{\zeta}) \right] \tag{16}$$

$$\Psi_{k,2}^{\zeta,(r)}(t, \boldsymbol{z}_k^{\zeta}) = \mathbb{E}_{q(\tau^{(r)})} \left[ \boldsymbol{\kappa}_k^{\zeta}(\boldsymbol{z}_k^{\zeta}, \tau^{(r)}(t)) \boldsymbol{\kappa}_k^{\zeta}(\tau^{(r)}(t), \boldsymbol{z}_k^{\zeta}) \right]$$