[Reviews · NeurIPS 2018]

Reviewer 1



This work extends Gaussian Process Factor Analysis (GPFA) by using a sparse variational GP approximation, leveraging inducing points to tackle the scalability issues with GPFA and allowing the use of a more realistic point process observational model. Furthermore, the authors propose a nested GP approach for simultaneous inference and temporal alignment of the latent trajectories across trials. The proposed extension are tested in a simulated dataset, and compared with other relevant methods. Finally, the authors apply the proposed method to neuronal recordings in motor cortex, showing not only that the temporal alignment leads to better cross-validated performance, but that the temporal warping inferred on a single trial basis is related to behavioural variables such as go cue and reaction time. Quality: The approach taken here is extremely interesting both technically and from a neuronal data analysis perspective. The testing provided is thorough and convincing, and the application to neuronal recordings elevates the quality of the submission even further. No qualms here, great work! Clarity: The presentation is very clear (which is impressive given the subject and the number of innovations introduced) and the paper is nicely structured. Highly polished paper. Originality: This work borrows from several recent advancements to propose a new GPFA extension. While there are a number of GPFA extensions where different noise models are considered I don’t know of any other method that also joins the temporal alignment across trials in this way. Significance: Not only are the extensions proposed technically interesting, they are of high interest for the analysis of neuronal recordings, particularly the effort to understand changes in brain state across trials. I believe this work can have a significant impact in the analysis of high dimensional neuronal activity. =============================================================== Nothing to add on my end regarding the authors' response. I maintain my score (8).

Reviewer 2



-- Paper Summary This paper tackles the problem of identifying patterns in spiking population activity of neurons by considering latent structure and trajectories. This is a particularly challenging problem due to potential time misalignment between multiple trials. Additionally, the various time series are generally assumed to be independent while there are also issues with tractability due to linear algebraic operations involving a TxT kernel matrix. This work tackles the latter issue by introducing the sparse variational approximation within the GPFA model, and addresses shared latent structure across trials (or even partitions of trials) by exploiting a nested GP architecture. -- Originality + Significance The variational sparse approximation to Gaussian process inference has become the go-to approach for solving tractability issues with GPs without compromising on performance. Although the approximation in itself has been used and applied extensively in other papers, to the best of my knowledge, this paper is the first in applying it to the GPFA model in particular. In combination with the point process likelihood being considered, the resulting model is sufficiently different from existing approximate models based on sparse VI while also incorporating additional elements (explored in Section 4 of the paper). A recent paper discussing simultaneous modelling of data and temporal alignment, ‘Gaussian Process Latent Variable Alignment Learning (Kazlauskaite et al)’, seems to be related in terms of scope. In fact, the generalisation to GPLVMs is listed in the concluding remarks of the paper under review. Although there is a clear distinction between the two papers, particularly with regards to the likelihood considered, any similarity to this work should be duly discussed. It might be interesting to draw a connection to papers such as ‘Collaborative Multi-output Gaussian processes (Nguyen & Bonilla)’, where inducing points are shared across multiple latent functions. It is not clear whether the inducing points considered in this paper are unique to each latent function or shared between them, although I believe it’s the former. If not, this could possibly contribute towards further improving scalability. -- Quality The model is clearly detailed in Sections 3 and 4. As highlighted earlier, the sparse variational approximation is fairly standard; however, the authors do not limit themselves to the base approximation, but also consider several extensions which make the paper more complete and well-rounded. Once all aspects of the model are described, I encourage the authors to summarise the overall complexity of the model with regards to how many trials, conditions, time points etc are considered in a given experiment. -- Writing and Clarity The paper is very well-written and a pleasure to read - I did not spot any typos while reading it. The figures are well-captioned and easy to interpret. Corresponding discussion of the figures in the text is also appropriate. Some minor comments: - While the ‘macaque’ species might be well-known in the biology domain, I would suggest adding a footnote or small parenthesis explaining the term first. It is currently introduced in L57 without much context. - Similarly, it would be good to explain what is ‘innovations noise’ (L198) - ‘Where’ in L121 could be changed to lower case The content provided in the supplementary material complements the main text with more detailed proofs and figures. I found the balance between the two to be just right. -- Overall Recommendation This is a well-written paper with clearly established goals that are both individually and jointly investigated in this paper. The contributions themselves are fairly modest and application-specific; nonetheless, state-of-the-art results are obtained in the problem setting being investigated, while the majority of the modelling choices are sufficiently general to be of interest to the broader NIPS community. ------------ -- Post-review Thank you for replying to my queries, in particular for expanding on the relation of this work to the indicated papers and detailing the overall complexity of the method. I believe that incorporating this discussion into the main paper (along with other minor fixes) will only elevate what is already a very good piece of work!

Reviewer 3



The authors propose a method for decomposing the responses of neural populations into low-dimensional structures. This paper improves upon previous Gaussian process Factor Analysis methods by using a variational approximation to perform inference in continuous time and with stochastic timing differences between trials. The methods were well presented and easy to follow. The motivation behind applying the timewarping methods to neural data is strong, and the variational approach makes this methodology practical for real-world applications. The results did not, however, demonstrate how these methods behave and scale as a function of neurons, experimental conditions, or trials. One small detail that I missed was how the strictly increasing property of the timewarping process was enforced.